# Hybrid Local and Global Deep-Learning Architecture for Salient-Object Detection

**Wajeeha Sultan [1], Nadeem Anjum [1,*,†] [ID], Mark Stansfield [2] and Naeem Ramzan [2,†]**

[1] Department of Computer Science, Capital University of Science and Technology, Islamabad Expressway, Kahuta Road Zone-V Sihala, Islamabad, Islamabad Capital Territory, Pakistan; wajeehasultan543@gmail.com

[2] School of Engineering and Computing, University of the West of Scotland, Technology Ave, Blantyre, Glasgow G72 0LH, UK; Mark.Stansfield@uws.ac.uk (M.S.); naeem.ramzan@uws.ac.uk (N.R.)

* Correspondence: nadeem.anjum@cust.edu.pk

† Senior Member, IEEE.

**Abstract:** Salient-object detection is a fundamental and the most challenging problem in computer vision. This paper focuses on the detection of salient objects, especially in low-contrast images. To this end, a hybrid deep-learning architecture is proposed where features are extracted on both the local and global level. These features are then integrated to extract the exact boundary of the object of interest in an image. Experimentation was performed on five standard datasets, and results were compared with state-of-the-art approaches. Both qualitative and quantitative analyses showed the robustness of the proposed architecture.

**Keywords:** deep-learning models; salient-object detection; hybrid architecture; boundary-aware refinements

## 1. Introduction

Human beings very easily and quickly make vision-based decisions by extracting all important information from any real scene. All important objects in a scene are extracted, as well as relevant useful characteristic information such an object's color, size, relative distance, and nearness. That is why salient-object detection (SOD) is extremely important and fundamental in many vision-related applications like computer vision, graphics, and robotics [1]. Due to its importance, much work is conducted in various research areas such as image captioning, target detection, scene classification, and semantic segmentation by utilising image-level annotations and image-quality evaluation [2–7].

Recently, deep learning has had remarkable success in salient-object detection because it provides a rich and discriminate representation of images. Convolutional neural networks (CNNs) are a very effective tool in machine learning, it working efficiently in salient-object detection because of the ability to extract both high- and low-level features. Early deep saliency models utilise multilayer perceptron (MLP) for detection. In these methods, the input image is split into small regions, and a CNN is then used to extract features that are passed to MLP to compute the saliency of the region. However, these models fail to extract high-level semantic information; thus, this semantic information is unavailable to further pass into fully connected layers. Therefore, this results in loss of information that is needed to obtain complete characteristic information of a salient object [1].

As an alternative, fully convolutional networks (FCNs) are often used [8]. In spite of the fact that FCNs produce good results, these models underperform for the images with low contrast, especially near boundaries of objects, because (1) saliency is heavily affected by the intense noise of low-contrast images; (2) in frequent pooling operations in deep-learning methods, loss of object structure and semantic information is unavoidable, which causes the poor detection of objects; and (3) since saliency

is determined by the global contrast of image instead of local features, it becomes hard for the model to examine detailed boundary knowledge of the object.

To overcome these problems, we propose a boundary-aware fully convolutional network for the detection of salient objects that captures both the local and global context with a built-in refinement module to achieve segmentation with fine boundaries. Integration of both local and global features helps our model to accurately locate salient region. Features produced from multiple convolutional layers are merged into an integrated feature map to obtain the global context.To capture the local context and enhance local contrast features, multiresolution feature maps resulting from convolutional layers, contrast features, and upsampled feature maps produced from deconvolution layers are merged to form the local feature map. The predicted saliency map is produced by the fusion of global and local features. The predicted map is then passed to the refinement module, which is built as a residual block that purifies the input saliency map (see Figure 1).

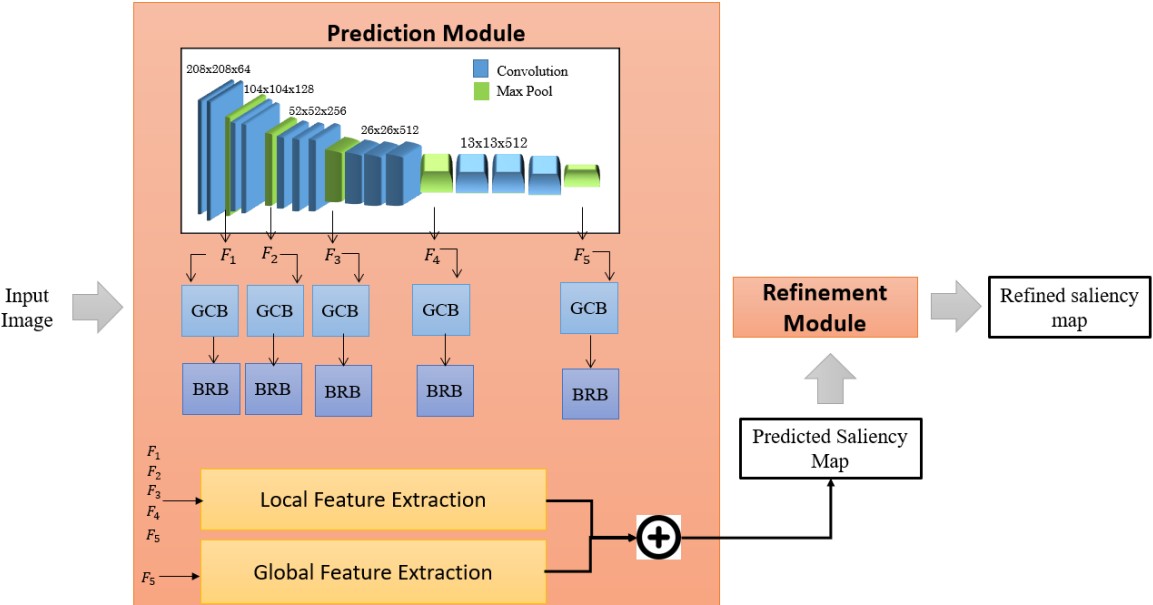

**Figure 1.** Overview of proposed model (SODL), which consists of prediction module and refinement module. Image is passed to prediction module that learns deep features and produces a predicted saliency map, which is refined by the refinement module. This refined saliency map is taken as the final map.

The contributions of this work are summarised as follows:

1. Generation of local and global deep feature maps to extract detailed structure of objects.
2. Enhanced detection of objects in low-contrast images by integrating contrast features, and a global convolutional modules to introduce dense connections between features and classifiers.
3. Boundary-refinement module to preserve boundary information present in initial layers.
4. A residual refinement module is embedded that processes the predicted saliency map to refine boundaries by learning residuals.

The rest of this paper is organised as follows: Section 2 details the literature review, Section 3 explains the architecture of the proposed approach, Section 4 presents the experimental results, and Section 5 draws the conclusions.

## 2. Related Work

In general, methods for the detection of salient objects can broadly be classified into two groups: traditional and deep-learning-based approaches.

### 2.1. Traditional Approaches

These approaches use low-level features (e.g., color, contrast) to differentiate salient objects [9,10]. These features are useful in simple scenarios, but they have limited capability to caapture objects in complex scenarios. A comprehensive survey is provided by Borji et al. in [1].

### 2.2. Deep-Learning-Based Techniques

In deep learning, a model learns directly from image to perform classification. Wang et al. [11] developed a technique that makes use of prior saliency-detection techniques. Repetitive architecture allows for this method to refine the saliency map by rectifying previous errors, which results in fine predictions. They introduced a pretraining strategy using segmentation data for network training, which helps in segmentation or successful training and allows for the network to grasp objects for saliency detection. They trained an FCN to estimate the nonlinear mapping of saliency values from raw pixels and neglect saliency priors; then, these saliency priors were combined with deeply learned features, which are passed for iterative refinement. In this module the whole network is propagated forward, which causes an increase in computational cost and memory usage. Vivek et al. [12] used many saliency-detection methods to improve the quality of saliency map. First, they generated initial saliency maps by selecting saliency models. Then, they integrated these maps to form a binary map after that final saliency map had been generated using integration logic. The combined binary map defines pixels in finer way than a single binary map does. By using these labels, the final saliency map is produced by using logic in which maximal and minimal saliency values are assigned to pixels. The efficiency of this method relies on the saliency-detection model being selected. Therefore, the method of selecting existing techniques plays a key role. This method performs well when selected methods can detect images, and fails when those methods cannot. Feng et al. [13] introduced a model to enhance segmentation results near boundaries. An attentive feedback module was proposed to produce fine boundaries. Features from encoder blocks are passed to decoders where the feedback module is applied for segmentation. This module learns the structure and performs segmentation. Further, they proposed boundary-enhancement loss to assist the feedback module. VGG 16 was used as a base model by modifying it into an encoder network, followed by perception module that uses a fully connected layer that leads to the decoder module. Every decoder layer had a $3 \times 3$ convolutional layer. Multiscale features are collected via the encoder block; then, global saliency prediction is calculated, which is passed to the decoder for finer saliency predictions. For boundary correction, attentive feedback is applied.

Wang et al. [14] proposed a deep model that uses boundary knowledge to accurately locate salient objects. In boundary-guided networks, two subnetworks are defined: one for the mask and the other for boundaries; features are shared between these two networks. They also proposed focal loss to learn the loss of hard pixels near boundaries. Both subnetworks follow encoder decoder networks, features are extracted in the encoder, and the decoder gives the output. Decoders of both subnetworks are connected. The features extracted by the encoder have different resolutions, and bidirectional flow of information is enabled in encoders. Feature maps from two subnetworks are combined and passed to a convolutional layer. The decoders of mask and boundary subnetworks are connected. The mask decoder refines the mask prediction and uses focal loss, which focuses on boundary pixels. Girshick et al. [15] developed a robust and versatile detection technique that enhances the mean average precision by over 50% compared to prior techniques. This approach incorporates two concepts: (i) applying high CNNs to localise objects from bottom–up regions; and (ii) supervision pertaining to when abundant data are available, followed by fine tuning to increase performance. The whole technique comprises three modules. First, the module originates the regions. The second module consists of convolutional layers to collect features from regions. The final module is linear SVMs. The authors used selective search to select the technique for the region proposals. To extract features, regions are first converted into a form that is compatible with CNN; after features are extracted, optimisation is applied. Eitel et al. [16], proposed an RGB-D strategy for object detection. The network

consisted of two different CNNs for each modality paired with a late fusion network. They set up a multistage training approach for effective training, and a data-augmentation framework for learning with images by manipulating them with noise patterns. The two convolutional models utilise color and depth information to examine RGB and depth data, which is integrated into a fusion technique. ImageNet is used for data pretraining; then, multimodal CNN is trained. Parameters are tuned for data classification, followed by the combined training of the parameters and fusion network. Images are resized before passing them to CNNs. The authors proposed an effective method for encoding depth images that normalises depth values that fall between 0 to 255. Then, the color amp is applied on images, which transforms them into three channel images. Table 1 highlights some of the other related state-of-the-art approaches available in the literature.

**Table 1.** Summary of related-work review.

| Ref | Methods | Strengths | Limitations |
|-----|---------|-----------|-------------|
| [10] | Frequency-tuned model for large salient objects | Efficiency in computation Fine detection in large objects | Alignment issues and poor performance for low contrast |
| [11] | FCN-based saliency map | Good saliency refinement | Computational iterative process |
| [12] | Hybrid saliency detection | improved accuracy | Poor performance for low-contrast images |
| [16] | Multistage training | Good noise filter | Computationally complex |
| [17] | Mean shift | Robust against low noise | Low results for highly noisy images |
| [18] | Intensity- and contrast-based | Fine saliency map | Not efficient for large images |
| [19] | Image signature | Scalability | Poor performance for small objects |
| [20] | Background construction for saliency | Better accuracy | High computational cost |
| [21] | Graph-based manifold ranking | Combination of multiple methods | High dimensionality and computational cost |
| [22] | Wavelets | effective saliency detection | Loss of spatial information |
| [23] | Unsupervised technique for saliency | Joint and iterative optimisation | Computational cost is high |
| [24] | Bootstrap | Time reduction | Limited results shown |
| [25] | Contextual info | Good results | Perform low for complex backgrounds |
| [26] | Hierarchical and SVM | Lower computational cost | Perform poor for complex images |
| [27] | Uses scene knowledge | Good precision | Poor results for low-resolution images |
| [28] | Blockwise scene scanning | Improved accuracy | Computational cost |
| [29] | Regions based features | Efficient approach | Perform poor for low contrast images |
| [30] | Low- and high-level features | Improved accuracy | Bad segmentation for low-contrast images |

Both traditional and deep-learning-based models perform well for images having good contrast; however, their performance substantially degrades when contrast is low, and objects of interest have blurred boundaries. This work detects salient objects in low-contrast images using hybrid deep-learning models.

## 3. Proposed Approach

### 3.1. Architecture Overview

The proposed method consists of two main modules: (a) prediction of saliency maps and (b) refinement of the predicted saliency module. The prediction module consists of fully convolutional network that captures both local and global features. It also contains global-convolution and boundary-refinement blocks that focus on the refinement of features and the better extraction of boundary features. To further emphasise low-contrast images and enhance contours, a contrast module is also embedded in the model. Details are provided in subsequent sections.

### 3.2. Prediction Module

In this work, the prediction module was based on fully convolutional layers. To this end, the entire model network was based on a convolution operation and a deconvolution layer with varying output dimensions. This enabled our model to capture global and local features from different resolutions, as shown in Figure 1.

An input image is passed to five convolutional blocks to produce five feature maps $F_1$, $F_2$, $F_3$, $F_4$, and $F_5$. Every convolutional block has kernel size of $3 \times 3$, followed by a max-pooling operation with stride 2 in order to reduce spatial resolution to $13 \times 13$ from $208 \times 208$. Pooling decreases the number of parameters and computations. Max pooling is obtained by using a max filter that captures the maximum of the selected region and produces a new one. After that, five global convolutional blocks are added into the network that allow for dense connection in the features and convolutional layers, which enables features to acquire diverse neural information and be immune to local disruptions. We also added boundary-refinement blocks (BRBs) for the feature maps to keep boundary information. These BRBs are based on a residual structure in which input to this block is directly added to the output of the deeper layer, which helps to maintain information present in the initial layers, and this information is needed by the upsampling layers. This structure also helps in avoiding information loss due to a multilayer network architecture.

To generate local feature maps $F_L$ , five more convolutional blocks were added to the network, which processes the output of first five convolutional blocks as shown in Figure 2. These convolutional layers also have a kernel of $3 \times 3$ with 128 channels. The resulting diverse-scale local feature maps were $F_6$ , $F_7$, $F_8$, $F_9$, $F_1 0$. To further capture contrast feature $F_i^c$ ($i = 6, ..., 10$) for each feature map, which is essential, especially for low-contrast images, we calculated the difference between feature map $F_i$ and its local average $F_i^*$. In order to calculate the value of $F_i^*$. We employed average pooling with a kernel size of $3 \times 3$. The contrast feature was thus calculated as

$$F_i^c = F_i - F_i^* \tag{1}$$

Since pooling operations were performed after convolutional layers, the spatial size of the image was decreased (to reduce computational cost). However, in the deconvolutional layer, we employed upsampling in order to attain the original image size. The final feature map was generated by the concatenation of contrast feature $F_i^c$, local feature $F_i$, and upsampled feature map $F_{i+1}^u$ using the equation below:

$$F_i^u = \lambda[\Gamma(F_i, F_i^c, F_{i+1}^u)], \tag{2}$$

where $\Gamma$ is the concatenation function and $\lambda$ is the upsampling function. To calculate final local map $F_L$ , local contrast feature $F_6^c$ , local feature $F_6$, and upsampled feature map $F_7^u$ are concatenated and passed to convolution layer $\Xi$ with kernel size set to $1 \times 1$.

$$F^L = \Xi(\Gamma(F_6, F_6^c, F_7^u)). \tag{3}$$

For the computation of global feature map $F_G$, global features were extracted before allocating the saliency information of small regions. For this purpose, three convolutional layers with kernel size $3 \times 3$ and dilation with scale factor 2 is applied. Dilated convolutions reduces the loss of resolution. Figure 3 shows the structure of global convolutional block. Each convolutional layer was accompanied by an RELU activation function and a boundary-refinement block. Figure 4b,c shows the local and global feature maps of an input image. To produce the predicted saliency map, global $F_G$ and local feature maps $F_L$ were combined, and the resulting saliency map $S_P$ was passed as input to the refinement module. Figure 4d shows the predicted saliency map produced by the combination of local and global feature maps.

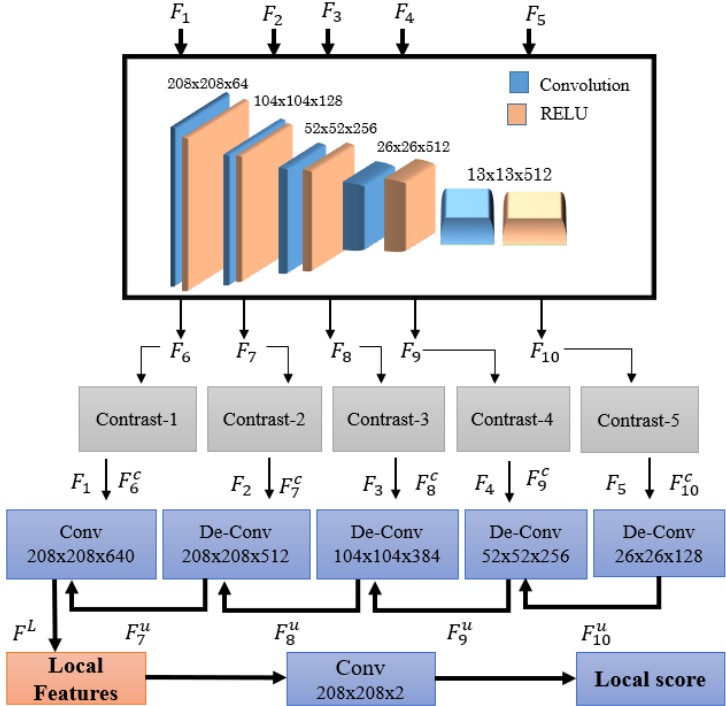

**Figure 2.** Extraction of local features. Local feature map produced by integration of contrast features, feature maps produced from convolutional layers, and upsampled feature maps.

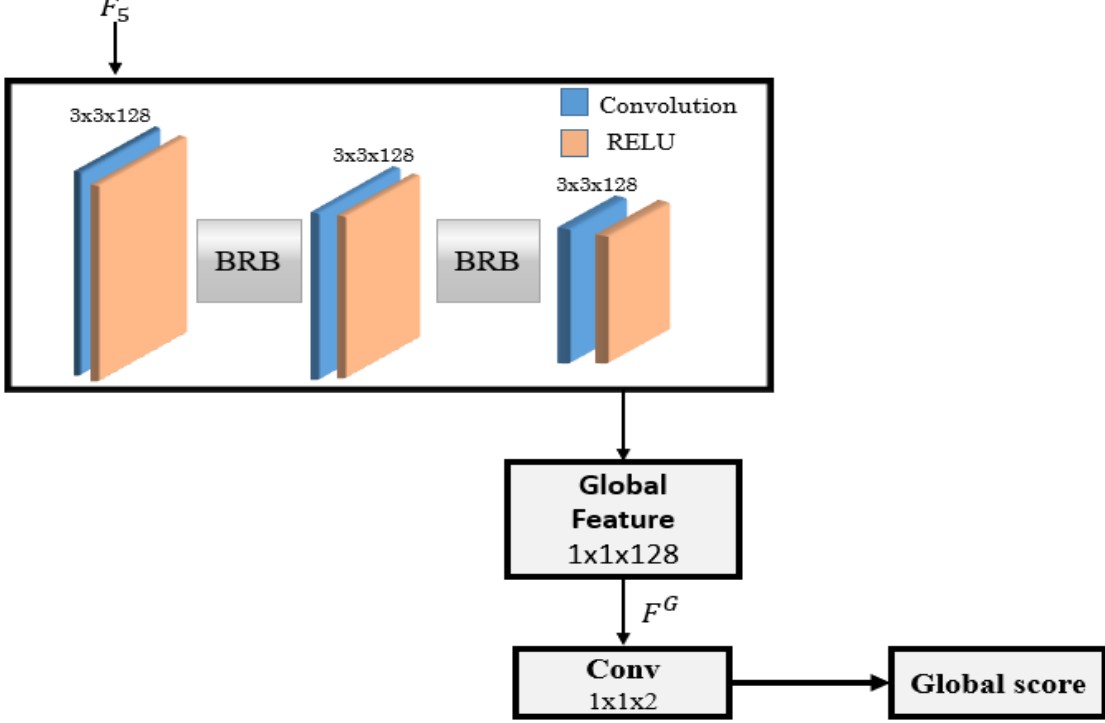

**Figure 3.** Extraction of global features. Feature map from fifth convolutional layer was used, passed through dilated convolutions. For maintaining spatial information, boundary-refinement blocks (BRBs) were also used.

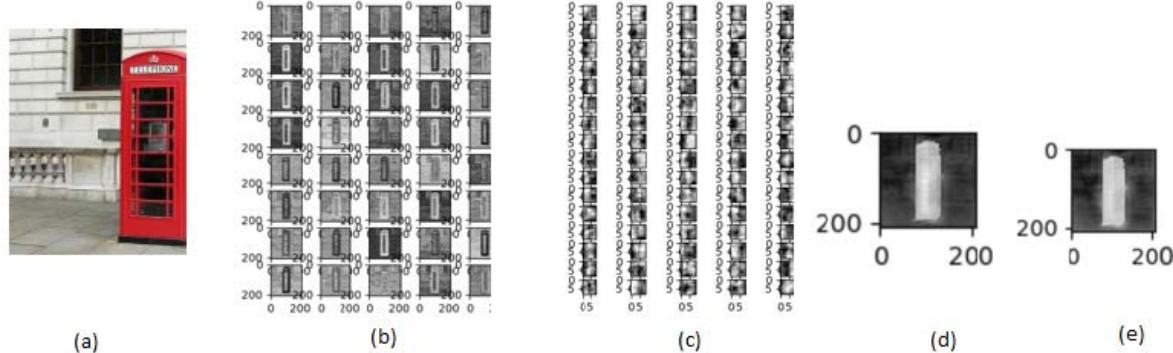

**Figure 4.** Output of various components of proposed approach: (**a**) input image, (**b**) local feature map, (**c**) global feature map, (**d**) predicted saliency map, and (**e**) refined saliency map.

### 3.3. Global Convolutional Block

Enhancement in classification was achieved by using a global convolutional block (GCB). This helps to efficiently exploit accessible visual information in low-contrast images. The GCB seeks to extend the receptive region of feature maps. It also introduces dense connections between features and classifiers, which helps the model to identify the salient region with very little additional computational cost.

As mentioned before, the GCB enhances the classification efficiency of proposed model by considering the dense connections between classifiers and feature maps, which enables the network to handle different types of transitions. GCB's large kernel is also useful in encoding more spatial information from feature maps, which increases precision in salient-object detection.

The GCB has two subdivisions that consist of two convolutional blocks. The left block has $7 \times 1$ convolutional layer, after which comes another convolutional layer of $1 \times 7$. The right block has $1 \times 7$ convolution followed by $7 \times 1$ convolution. These two subdivisions are combined to allow for dense connections, which enhances the receptive field's validity. GCB's computational cost is fairly low, which makes it more practicable.

### 3.4. Boundary-Refinement Block

The second module of the proposed network is the boundary-refinement block (BRB). It is a residual framework implanted after first five convolutional layers to maintain the boundary information of objects. This block is added for the refinement of feature maps and to enhance the accuracy of an object's spatial location. The BRB was included to improve localisation near boundaries, which can significantly improve object detection in low-contrast images.

### 3.5. Refinement Module

The refinement module refines the overall saliency map. This module is different from BRB because the latter is used to refine the feature maps and preserve the rich spatial information present in initial layers, while the refinement module refines the predicted saliency map to enhance model accuracy. Predicted saliency map $S_P$ is input for this refinement module. Typically, a refinement module is built as a residual block that purifies the input saliency map through learning residuals between saliency map and its ground truth.

A refinement module is embedded to improve both regional and boundary limitations in coarse saliency maps. Coarse saliency maps refer to boundary blurriness and the uneven prediction of probabilities in regions. The refinement module consists of an input layer followed by an encoder, a decoder, a link stage between encoder and decoder networks, and an output layer. Both encoder and decoder networks comprise four stages. Every stage consists of one convolutional layer with 64 filters of $3 \times 3$ size. After each convolutional layer, batch normalisation and RELU function are

performed. The link stage also contains a convolutional layer with the same filter and size, after which normalisation and RELU are performed.

To downsample in an encoder, max pooling is used; for upsampling, bilinear interpolation is used in the decoder. The resulting map is final saliency map $S^M$. Final saliency map $S_M$ is produced by the refinement of the predicted saliency map. The soft-max function computes the probability that a pixel p in a feature map belongs to a salient object:

$$S^M(p) = P(G^T(p) = l) = \frac{e^{w_l^L F^L(p) + v_l^L + w_l^G F^G + v_l^G}}{\sum_{l' \epsilon (0,1)} e^{w_{l'}^L F^L(p) + v_{l'}^L + w_{l'}^G F^G + v_{l'}^G}} \qquad (4)$$

where ( $W^L$ , $v^L$ ) and ( $W^G$ , $V^G$ ) are linear operators, and $G^T$ is the ground truth. The loss function is the combination of cross-entropy loss and boundary loss. Figure 4e shows a refined saliency map. The results of the proposed approach on low-contrast images are compiled in Figure 5. The steps involved in encoding and decoding are shown in Figure 6.

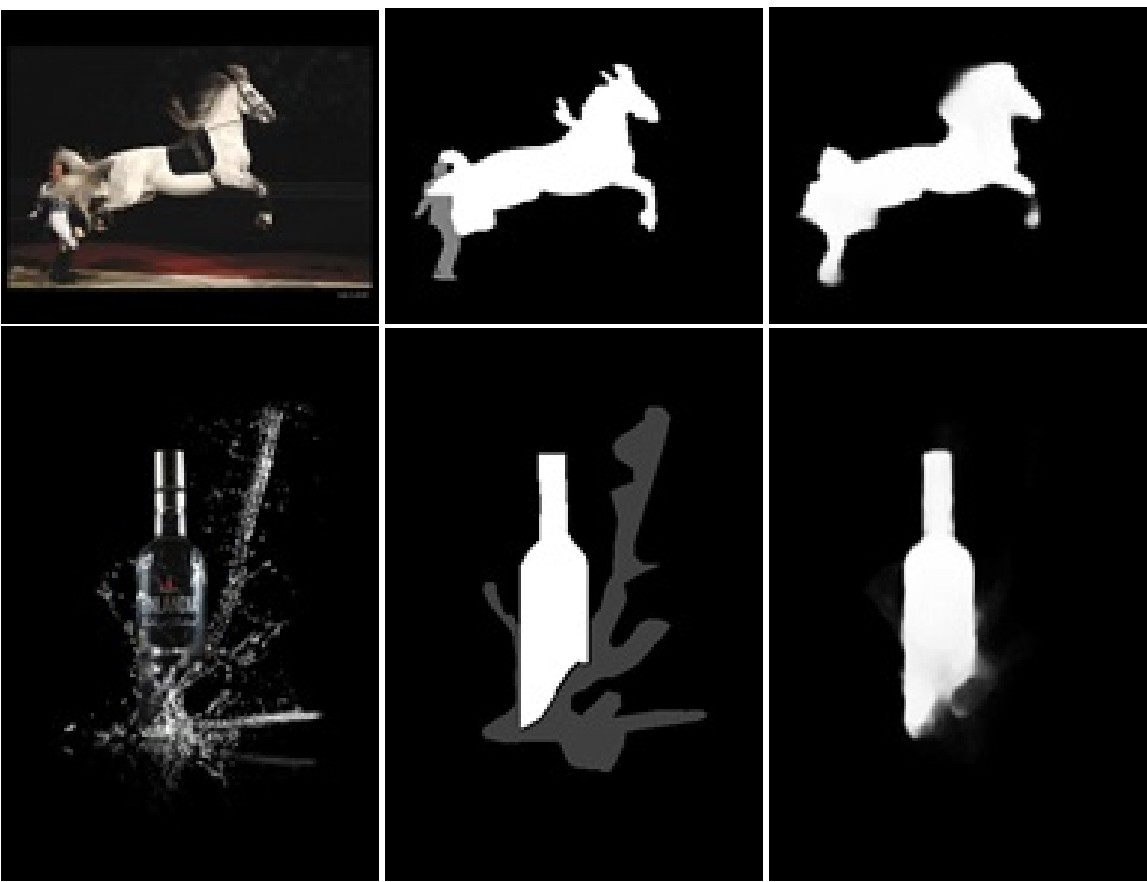

**Figure 5.** Salient-object detection results of proposed approach on low-contrast images: (**Column 1**) input; (**Column 2**) ground truth; (**Column 3**) proposed-approach results.

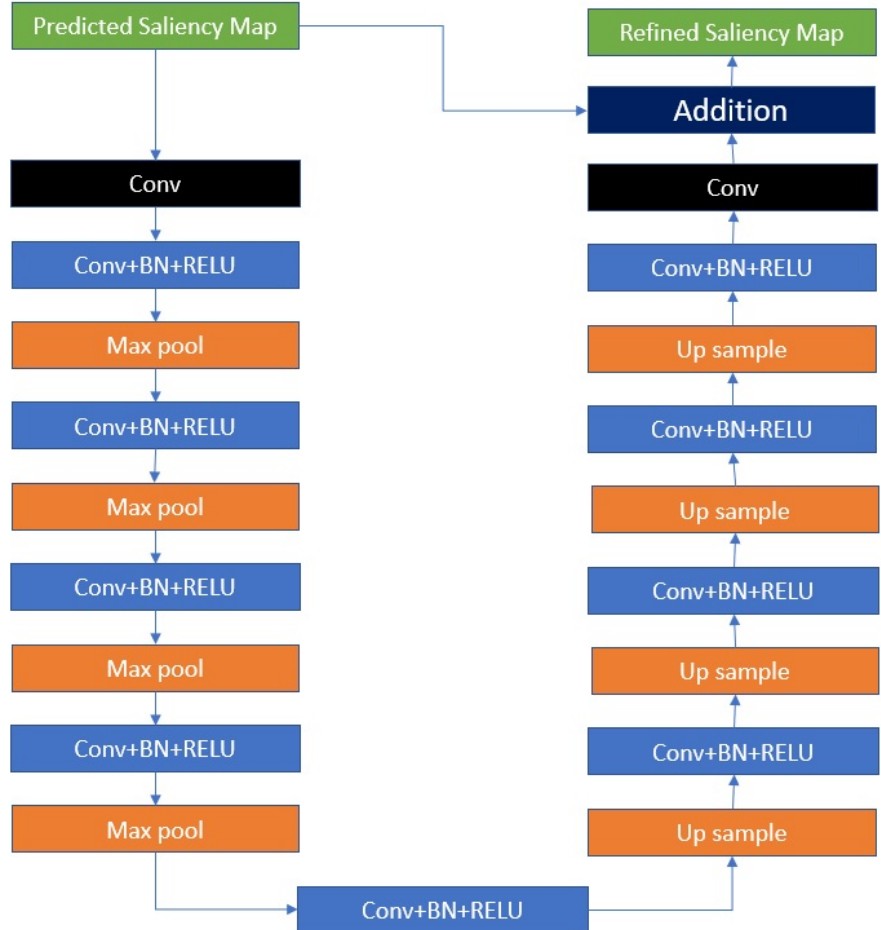

**Figure 6.** Encoder/decoder flow to extract refined saliency map.

## 4. Experimental Results

### 4.1. Datasets for Evaluation

Testing of the proposed approach was performed on five benchmark datasets:

- MSRA-B [31] comprises 5000 images and contains a single object mostly around the center position along with the bounding-box label.
- DUT-OMRON [32] includes 5168 images with complex backgrounds and a variety of content. Pixelwise ground-truth annotations are also available.
- The PASCAL-S [33] dataset contains 850 complex images. Eye-fixation records, and nonbinary and pixelwise annotations are also available.
- HKU-IS [34] comprises 4447 multiple distant objects. In these, a minimum of one object is present at the image boundary. Less difference between background and foreground makes these images more complex.
- DUTS [35] is very large Sod dataset containing 5019 test images and 10,553 training images. Many models use this dataset for training.

### 4.2. Evaluation Metrics

The following evaluation measures were used to measure the efficiency of the proposed model with other models:

- Precision–recall (PR) curve: Calculated by the conversion of the predicted saliency map into a binarised map and ground truth. Thresholding of 0–255 is applied to produce the binary map.

For all saliency maps present in a dataset, every binarising threshold comes in a set of average precision and recall. An order of precision–recall pair is generated when the threshold varies from 0 to 1, which is used to plot the PR curve.

- F-measure curve: To provide comprehensive analysis, $F_\beta$ is calculated on both precision and recall as:

$$F_\beta = \frac{(1 + \beta^2)Precision \times Recall}{\beta^2 Precision + Recall} \qquad (5)$$

The $\beta$ value is set as 0.3 to highlight precision more because the rate of recall is not as significant as that of precision. The value of the average F-measure is also presented in this research. The F-measure curve is generated with a comparison of the binary map with the ground truth that is obtained by changing threshold to decide if a pixel is owned by a salient object.

- Mean absolute error (MAE) is used to correctly measure false-negative pixels. It calculates the pixelwise error between saliency map and ground truth:

$$MAE = mean(|S^M - G^T|) \qquad (6)$$

If the MAE value is less, it indicates that ground truth $G^T$ and predicted saliency map $S^M$ are highly similar.

### 4.3. Implementation and Experiment Setup

The MSRA-B [31] dataset was used to train the model, and the training set contained 2500 images and 500 images for validation. To avoid overfitting, the horizontal flipping of images was used.

Pretrained VGG16 [36] was used for feature extraction. When passed to the proposed approach, the image was resized to $416 \times 416$. An Adam optimiser was utilised for training, and parameters were set to the default values.

Publicly available framework TensorFlow was used to implement this model. An Nvidia GeForce GTX 1080 (8119 MiB) GPU with 62 GB RAM and a Linux operating system was used to train the model, while model testing on the different datasets was performed on a Core i5 CPU with 8 GB RAM and a Windows operating system.

### 4.4. Comparison with the State of the Art

The proposed model was compared with four state-of-the-art saliency-detection models on five datasets. The deep saliency models included nonlocal features (NLDF) [29], contour to saliency (C2S) [11], visual saliency detection based on multiscale deep CNN features (MDF) [37], deep saliency with encoded low-level distance map and high-level features (ELD) [30], salient-object detection in low-contrast images via global convolution and boundary refinement (GCBR) [38], and aggregating multilevel convolutional features for salient-object detection (Amulet) [39].

#### 4.4.1. Quantitative Comparison

To determine the accuracy of the objects that the proposed model segmented, the proposed model and four others were tested on five highly used datasets. To show the results, we plotted precision, recall, and F-measure curves on each dataset (see Figure 7). Additionally, results of the proposed model and the four others in terms of mean absolute error (MAE) and weighted F-measure (WF) are presented (see Table 2).

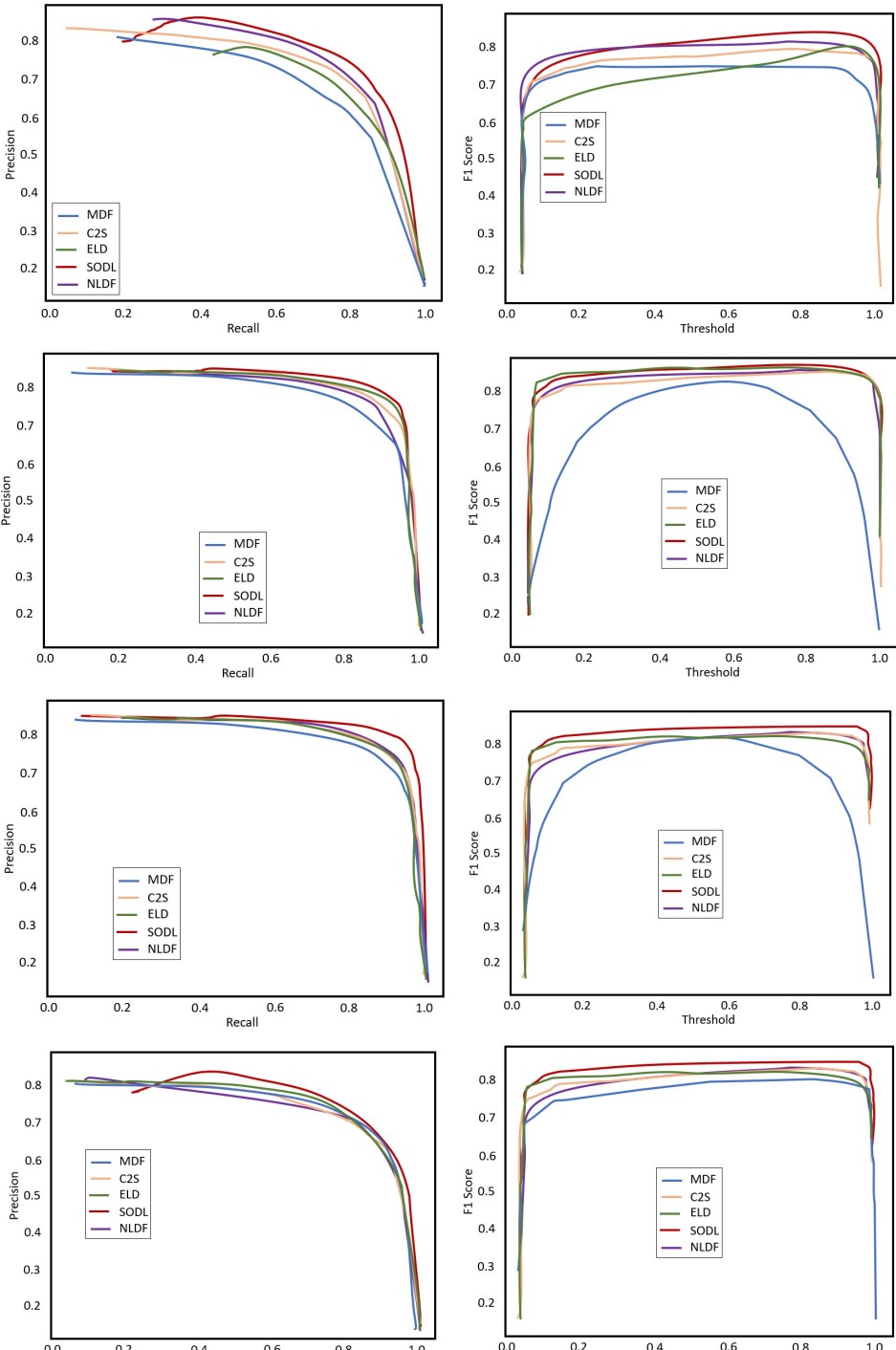

**Figure 7.** *Cont.*

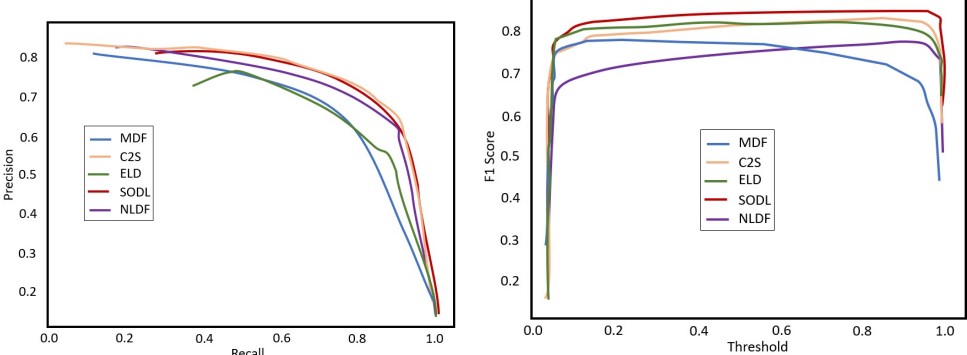

**Figure 7.** (**left**) Precision–recall (PR) and (**right**) F-measure curves of proposed model (SODL) and state-of-the-art (SOA) models.

**Table 2.** Comparison of proposed method with SOA methods on five datasets in terms of F-measure (larger is better) and MAE (smaller is better). Note: NLDF, nonlocal features; C2S, contour to saliency; MDF, multiscale deep CNN features; ELD, deep saliency with encoded low-level distance map and high-level features; GCBR, salient-object detection in low-contrast images via global convolution and boundary refinement; Amulet, aggregating multilevel convolutional features for salient-object detection.

| Dataset | Criteria | Proposed | NLDF | C2S | MDF | ELD | GCBR | Amulet |
|---------|----------|----------|------|-----|-----|-----|------|--------|
| MSRA-B | MAE | **0.0345** | 0.0477 | 0.0662 | 0.104 | - | 0.0373 | - |
|  | $F_\beta$ | **0.935** | 0.910 | 0.8309 | 0.885 | - | 0.8904 | - |
| DUTS | MAE | **0.0653** | 0.066 | 0.0663 | 0.094 | 0.093 | 0.0695 | 0.075 |
|  | $F_\beta$ | **0.816** | 0.812 | 0.790 | 0.730 | 0.738 | 0.801 | 0.773 |
| DUT-OMRON | MAE | **0.0753** | 0.0795 | 0.0790 | 0.0915 | 0.0909 | 0.0763 | 0.0830 |
|  | $F_\beta$ | **0.764** | 0.753 | 0.733 | 0.694 | 0.719 | 0.7010 | 0.7370 |
| PASCAL-S | MAE | 0.106 | 0.113 | 0.0991 | 0.143 | 0.133 | **0.0356** | 0.092 |
|  | $F_\beta$ | **0.829** | 0.807 | 0.827 | 0.771 | 0.768 | 0.801 | 0.826 |
| HKU-IS | MAE | **0.0432** | 0.0485 | 0.0527 | 0.135 | 0.074 | **0.0432** | 0.052 |
|  | $F_\beta$ | **0.918** | 0.914 | 0.897 | 0.867 | 0.839 | 0.8988 | 0.889 |

Figure 7 shows the PR and F-measure curves of the proposed model with four other state-of-the-art models, which show that the proposed model outperformed all other methods. The other quantitative comparison in terms of weighted F-measure and mean absolute error in Table 2 showed that the proposed model was first among all other models, and the F-measure was improved by 10%, 8%, 7%, 6.1%, and 7.9% for the MSRA-B, DUTS, DUT-OMRON, PASCAL-S, and HKU-IS datasets, respectively.

### 4.4.2. Qualitative Comparison

To further highlight the performance of the proposed model and quality of tge saliency map, qualitative analysis of the proposed model on five datasets was also performed. Figure 8 shows the quality of the maps and segmented salient objects from five widely used datasets.

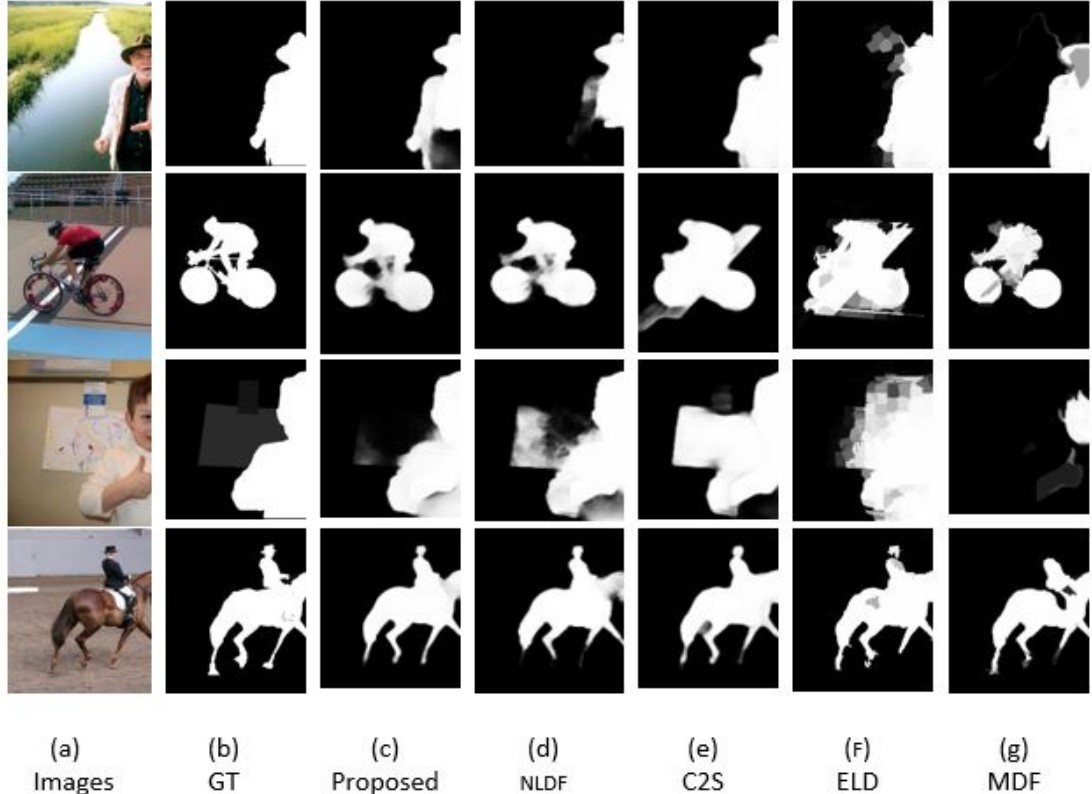

**Figure 8.** Visual saliency maps generated by our method and four other methods. Ours achieved the best results, especially in recovering spatial details of salient objects.

Qualitative comparison of the proposed model on the five datasets showed that the proposed model could accurately segment objects in different challenging scenarios, containing images that had much less color difference from their background, and objects that were large and touching image boundaries. Most strategies can produce good results for images with simple scenarios, while the proposed model could achieve better results even in complex scenarios. Many deep-learning models cannot segment and locate objects in complex backgrounds, but the proposed technique was successful in capturing most salient regions.

## 5. Conclusions

This paper presented a deep convolutional network with the integration of local and global features with boundary refinement. The combination of these features helps in the accurate detection and segmentation of salient objects. The embedded global convolutional and boundary-refinement blocks help in better feature extraction and preserve spatial location present in initial layers, which are later refined by the refinement module, resulting in more distinct features and accurate detection. Moreover, we examined the capabilities of the proposed model on five large and widely used datasets. Experiment results indicated that the proposed method outperformed state-of-the-art methods and has high capability for many computer-vision tasks.

**Author Contributions:** "Conceptualisation": W.S., N.A. and N.R.; "Methodology": W.S. and N.A.; "Software": W.S.; "Validation": W.S., N.A. and M.S.; "Formal analysis": W.S. and N.A.; "Resources": N.A., M.S. and N.R.; "Initial draft": W.S. and N.A.; "Review and editing": N.A., M.S. and N.R.; "Supervision": N.A.; "Project administration": N.A. and N.R.; "Funding acquisition": N.R. All authors have read and agreed to the published version of the manuscript.

**Funding:** This work is partially supported by Under Erasmus$^{+}$ CBHE Project Number 619483.

**Conflicts of Interest:** The authors declare no conflict of interest.

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
