# Peer review of "Hybrid Local and Global Deep-Learning Architecture for Salient-Object Detection"

_applsci, doi:10.3390/app10238754_

Round 1

Reviewer 1 Report

The paper presents a deep learning approach with local and global features for salient object detection. The overall work is presented in a clear manner with background literature, explanations of the method, and sufficient comparisons. Minor comments:

- Please go through the text for minor typos.
- Some references are not numbered in the main text, e.g. line 106.

Reviewer 2 Report

  1. The English writing of this article needs to be revised carefully. There are many grammatical errors in this paper.
  2. One of the main contributions presented in this work is that the enhanced performance of salient object detection in low-contrast images. Please provide some experiments or examples to support this claim.
  3. Please check the correctness of eq. (1). The letter i seems to be the subscript of capital F. Besides, in this equation, how to calculate the local average F*?
  4. There are some strange operations in eqs. (2) and (3). For example, the function Upsample[], CAT(), and Conv(). Some of them can be achieved by layers of a network, and some of them need to be pre-defined by a mathematical operation. It is suggested to describe them in details.
  5. In lines 148-149, the sentence “three convolutional layers… and 512 dilations (where dilation =2) is applied”. Why 512 dilations are used here?
  6. In line 190, “The decoder follows same structure as encoder.” Is it correct? In general, the structure between them is symmetrical, not the same.
  7. Please improve the quality of figure 5. It is hard to observe.
  8. Please check table 2. The last two columns seem incorrect. Besides, why using the metric weighted F measure (WF) in this numerical comparison?

Generally, this paper proposes a deep-learning-based model to detect the salient object region pixelwisely. The results seems interesting, and worthy for publishing. However, the quality does not meet the standard of a periodical. Please improve this manuscript.

Round 2

Reviewer 2 Report

This version is much better than the previous one. Most parts of this paper are fine and seem having certain contribution of salient object detection. However, the quality of Figs. 6 and 7 are blurred. Please reproduce the figures. 
